# Existing guidance on reporting of consensus methodology: a systematic review to inform ACCORD guideline development

Esther J van Zuuren ,[1] Patricia Logullo,[2] Amy Price,[3,4] Zbys Fedorowicz,[5] Ellen L Hughes,[6] William T Gattrell[7]

For numbered affiliations see end of article.

**Correspondence to**
Dr Esther J van Zuuren;
e.j.van_zuuren@lumc.nl

## ABSTRACT

**Objective** To identify evidence on the reporting quality of consensus methodology and to select potential checklist items for the ACcurate COnsensus Reporting Document (ACCORD) project to develop a consensus reporting guideline.

**Design** Systematic review.

**Data sources** Embase, MEDLINE, Web of Science, PubMed, Cochrane Library, Emcare, Academic Search Premier and PsycINFO from inception until 7 January 2022.

**Eligibility criteria** Studies, reviews and published guidance addressing the reporting quality of consensus methodology for improvement of health outcomes in biomedicine or clinical practice. Reports of studies using or describing consensus methods but not commenting on their reporting quality were excluded. No language restrictions were applied.

**Data extraction and synthesis** Screening and data extraction of eligible studies were carried out independently by two authors. Reporting quality items addressed by the studies were synthesised narratively.

**Results** Eighteen studies were included: five systematic reviews, four narrative reviews, three research papers, three conference abstracts, two research guidance papers and one protocol. The majority of studies indicated that the quality of reporting of consensus methodology could be improved. Commonly addressed items were: consensus panel composition; definition of consensus and the threshold for achieving consensus. Items least addressed were: public patient involvement (PPI); the role of the steering committee, chair, cochair; conflict of interest of panellists and funding. Data extracted from included studies revealed additional items that were not captured in the data extraction form such as justification of deviation from the protocol or incentives to encourage panellist response.

**Conclusion** The results of this systematic review confirmed the need for a reporting checklist for consensus methodology and provided a range of potential checklist items to report. The next step in the ACCORD project builds on this systematic review and focuses on reaching consensus on these items to develop the reporting guideline.

**Protocol registration** https://osf.io/2rzm9.

## STRENGTHS AND LIMITATIONS OF THIS STUDY

⇒ This systematic review used a comprehensive search of multiple databases without language restriction.

⇒ The included studies ranged from conference abstracts and protocols to guidelines and systematic reviews.

⇒ For full transparency and to promote discussion, all data retrieved are reported.

⇒ The data extraction form used may have missed a few potential reporting topics, but these will be recovered, in the following stages of the ACcurate COnsensus Reporting Document project, by additional reviews and the Delphi panel experience.

⇒ Conclusions are limited by the paucity of studies that provided substantial useful guidance.

## INTRODUCTION

Healthcare providers face continuing challenges in making treatment decisions, particularly where available information on a clinical topic is limited, contradictory or non-existent. In such situations, alternative and complementary approaches underpinned by collective judgement and based on expert consensus may be used.[1–3]

A variety of approaches with differing methodological rigour can be used to achieve consensus-based decisions. These range from informal 'expert consensus meetings' to structured or systematic approaches such as the Delphi method and the Nominal Group Technique (NGT). These methods can be used for generating ideas or determining priorities and aim to achieve consensus through voting on a series of multiple-choice questions.[4–7] The voting process varies according to the method and may take place anonymously (as in Delphi) and/or face to face (in NGT and consensus conferences).[8–10] Key elements in the process include the use of valid and reliable methods to reach consensus and

subsequently their transparent reporting; however, these aspects are seldom clearly and explicitly reported.[3 11]

Reporting guidelines have been developed and are in use for the majority of study designs, for example, Preferred Reporting Items for Systematic Reviews and Meta-Analyses (PRISMA), Consolidated Standards of Reporting Trials (CONSORT) and Strengthening the Reporting of Observational Studies in Epidemiology (STROBE) (for all existing reporting guidelines, see: https://www.equator-network.org/). However, no research reporting guideline exists for studies involving consensus methodology other than best practice guidance for Delphi studies in palliative care.[12] Guidelines should include 'a checklist, flow diagram or explicit text to guide authors in reporting a specific type of research, developed using explicit methodology'.[3]

Deficiencies in the reporting of consensus methods have been well documented in the literature and are referred to in the protocol for the ACcurate COnsensus Reporting Document (ACCORD) project, which aims to develop a reporting guideline for methods used to reach consensus.[13] In accordance with the EQUATOR Network guidance in the toolkit for the development of reporting guidelines, the next step for the ACCORD project was a review of the relevant literature, which would ultimately inform the voting process.[3]

Our objective was to undertake a thorough and comprehensive systematic review that seeks to identify evidence on the quality of reporting of consensus methodology, for subsequent development into a draft checklist of items for the ACCORD guideline. This ACCORD reporting guideline will assist the biomedical research and clinical practice community to describe the methods used to reach consensus in a complete, transparent and consistent manner.

## METHODS
This manuscript conforms to the PRISMA statement[14] and follows a prespecified protocol.[13] The protocol was registered on 12 October 2021 at the Open Science Framework.[15]

### Inclusion criteria
Eligible studies consisted of reviews and published guidance, which addressed the reporting quality of consensus methodology and aimed to improve health outcomes in biomedicine or clinical practice.

### Exclusion criteria
Excluded were publications using consensus methods or describing consensus methods or discussing the advantages or disadvantages of frameworks, procedures or techniques to reach consensus, without specifically addressing reporting quality. Examples include guidelines developed through the use of consensus methodologies, such as reporting guidelines, clinical practice guidelines or core outcome set development studies. Editorials (usually brief opinion-based comments), letters about individual publications and commentaries on consensus methods outside the scope of biomedical research (eg, in the social sciences, economy, politics or marketing) were also excluded for this systematic review.

### Literature search strategy and data sources
A systematic literature search was conducted on 7 January 2022 by a biomedical information specialist. The following bibliographical databases were searched: MEDLINE (OVID version), Embase (OVID version), PubMed, Web of Science, MEDLINE (Web of Science), Cochrane Library, Emcare (OVID version), PsycINFO (EbscoHOST version) and Academic Search Premier. The full search strategy is presented in online supplemental material 1.

We (EJvZ, ZF, PL and WTG) piloted four initial search strategies provided by the information specialist (JWS, see Acknowledgements section). The initial search strategy was sensitive and precise, producing the highest number of retrieved references (N=7951). After several rounds of checking through known relevant references and controlling for the effect of the performance of certain search terms, modifications were made, including the use of the most explicit terms in the most specific search fields. The performance of search terms was investigated from two vantage points: homonymy (same search term, but different meaning), and, particularly, loss-of-context (right meaning of the word, but not in the correct context). This extended search strategy not only provided extra 'signal' but also reduced the level of 'noise'. We chose to use specific rather than broad terms (eg, not using the singular terms 'delphi' and 'consensus' instead we included these words with relevant phrases or with other contextual words). In this way, the refined search strategy was better aligned with our inclusion criteria and the objectives of the systematic review.

### Selection process
The final search results were uploaded to Rayyan (https://rayyan.ai) in the blind mode for independent screening by four review authors (EJvZ, ZF, PL and WTG) based on titles and abstracts. No language restrictions were applied. Records deemed eligible or without sufficient detail to make a clear judgement, we retrieved as full-text articles (EJvZ). The same four reviewers independently reassessed the eligibility of these full-text papers and any discrepancies were resolved through discussion. The references of the included studies were also checked for additional potentially eligible studies (EJvZ).

### Data extraction, collection of items and synthesis
Study details and outcome data from the included studies were collected independently within Covidence (https://www.covidence.org/) by two authors using a piloted data extraction form (EJvZ and WTG). The data extraction form questions were compiled based on the review authors' own experiences with reporting quality evaluation and literature on consensus methodology.

van Zuuren EJ, *et al. BMJ Open* 2022;**12**:e065154. doi:10.1136/bmjopen-2022-065154

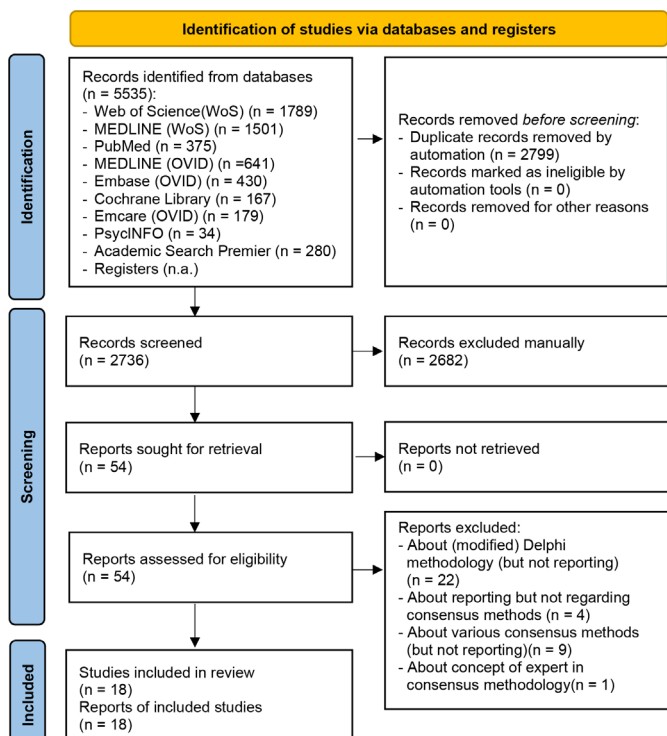

**Figure 1** PRISMA 2020 flow diagram for new systematic reviews, including searches of databases, registers and other sources.[14]. PRISMA, Preferred Reporting Items for Systematic Reviews and Meta-Analyses.

Furthermore, two additional free text fields were created for extractors to present issues addressed by the included studies that were not captured by the other questions, and for others that the extractors felt were not directly addressed by the studies but were rather inferences about topics that could be potential issues in the reporting of consensus methods. Disagreements were discussed and reconciled by consultation with a third and fourth author (ZF and AP).

The following details were extracted: bibliographic details and reporting items including any suggestions and comments regarding reporting items. Reporting items were divided into the component parts of background, methods, results and discussion, each addressing key aspects of consensus methodology. We also included a section for additional items retrieved from the studies and not captured in the data extraction form. The complete data extraction form is found as online supplemental material 2.

The topics extracted and the methods used in the studies included are synthesised narratively, in text and tables and online supplemental material. No further analyses were carried out, but these will follow during the next stage of the ACCORD project as per protocol.[13]

### Patient and public involvement

We involved patients, advocates and members of the lay public in the initial phases of this protocol,[13 15] as collaborators to develop this project and to coproduce the systematic review and coauthor the manuscript. They are collaborating with us by offering their experience with the use of consensus methods to develop guidelines and also systematic reviews. These contributors will work with us to disseminate the results.

### RESULTS

Our searches across the databases identified 2599 articles and 137 further references to abstracts totalling 2736 references (after removal of duplicates) (see figure 1). A total of 2682 records were excluded after examination of titles and abstracts. Full-text copies of 54 studies were obtained for further assessment of eligibility, and finally, just 18 eligible studies were included. Checking of the references of these full-text publications did not yield any additional eligible articles.

### Characteristics of included studies

Eighteen studies matched our prespecified eligibility criteria and were finally included in this review. These studies comprised five systematic reviews,[12 16–19] four reviews,[20–23] three research papers,[24–26] two research guidelines/guidance,[27 28] three conference abstracts[29–31] and one protocol.[32] Of the 18 included studies, 4 used Delphi plus other consensus methods[19 21 23 28] and the remaining 14 were primarily focused on only the Delphi method.[12 16–20 22 24–27 29 30]

### Characteristics of excluded studies

A total of 36 studies were excluded.[7 8 33–66] The main reasons for their exclusion were: that they discussed (modified) Delphi methodology but did not include aspects of reporting[33–54]; that they covered reporting but not on consensus methodology[55–58]; that various other consensus methodologies were discussed but not their reporting[7 8 59–65] and that only the concept of experts in consensus methodology was discussed.[66]

### Data extraction and narrative synthesis

The majority of studies indicated that reporting of consensus methods could be improved overall. The authors of these studies summarised some current limitations in reporting or proposed suggestions for improvement. Often there were common generic comments that noted reporting of consensus methodologies is inconsistent or lacks transparency. The studies provided few examples of areas that could be reported in more detail, such as: selection criteria for the participants and information about the participants; background information for panellists; definition of consensus; response rates after each round; description of level of anonymity or how anonymity was maintained and feedback between rounds (see table 1).

The studies we reviewed did not provide a systematic or standardised evaluation of the quality of reporting, but they did evaluate the literature critically and offered insights into the gaps of information about consensus. Fifteen papers made recommendations sometimes in the

**Table 1** Data on reporting quality of consensus methodologies

| Items that are not or not adequately reported in sufficient detail | |
|---|---|
| Selection criteria for participants/information about the participants[16 19 23 26 32] | Statement that anonymity was maintained or level of anonymity[20 21 25 28 29 32] |
| Literature review[20 21 31] | Type of consensus method used[29] |
| Background information for participants[20 21 25 28] | Threshold of consensus[29] |
| Recruitment strategies[19 22] | How questionnaire was developed[26] |
| Criteria for number of rounds[16 26] | Pretesting of instruments[19 32] |
| Stopping criteria[16 32] | Analysis procedure[24 32] |
| Feedback after rounds[17 20 21 23 25 26 28 31 32] | Changes to registered pre-analysis plan[24] |
| Rating scales used[31] | Reporting final number of list of items[32] |
| Criteria for dropping items[26] | Conflict of interest of panellists[29] |
| Response rates for each round[17 20 21 25 26 28 32] | Funding source[29] |
| Definition of consensus[17–19 21 23 25 26 28] | External support[29] |
| Level of consensus reached[19 31] | Generic comments that reporting needs improvement[12 17 26 30] |

form of short lists—based solely on the authors' opinion, rather than using a systematic approach to reporting guidance development.[12 16–25 27 28 30 32] Detailed statements regarding quality of reporting are reproduced in online supplemental material 3 .

In table 2, we summarise the results of the data extraction, which correlates the corresponding aspects of consensus reporting ('items') to the studies that address them. The items in the table are presented in the format used in the data extraction form (see online supplemental material 2).

The most frequently addressed item in the included studies (16 times) was the composition of and the criteria for selecting the panellists, including their demographics; specifically, age, gender, specialty, years of experience and sociodemographic background. The aspects of clarity in, and the importance of, defining consensus and the corresponding thresholds to reach that consensus were addressed in 13 studies. The prespecified number of voting rounds and provision of feedback to the panellists at the end of each round were addressed in 10 and 11 of the studies, respectively.

None of the included studies reported or made reference to public patient involvement (PPI). The roles of the steering committee/chair/cochair were not defined in any of the included studies. Reporting of the time interval between voting rounds, panel members' conflicts of interest (COI) and funding sources, as well as the measures used to avoid the influence of COI on voting and decision-making, were minimally addressed.

Conversely, three studies addressed between 12 and 19 reporting items of the 30 items present in the data extraction form of this review,[12 19 28] whereas two studies covered only two or three items.[19 24] We identified a considerable number of other aspects of reporting that were proposed in the included studies, but which were not captured in our data extraction form. These included: 'justifications for deviating from the protocol', 'incentives

for encouraging panellists to respond' and 'suggestions to add a flowchart of the consensus process'. All extracted data are found in online supplemental materials 4,5.

## DISCUSSION

Although consensus methodology is widely used in healthcare and researchers do raise poor reporting as an issue, we were able to identify only 18 studies that commented on reporting quality and/or provided suggestions to improve the quality of reporting of consensus methodology. These included studies ranged from conference abstracts and protocols to guidelines and systematic reviews. Only four studies covered methods other than the Delphi method and, thus, providing very limited guidance on other consensus methodologies. We carried out a comprehensive search of the most commonly used databases for systematic reviews without language restriction. However, during peer review of the present manuscript, three studies were brought to our attention as potentially eligible for inclusion.[67–69] Two of the studies had been excluded at the screening stage.[67 68] After full-text evaluation, one of the articles did discuss reporting quality but failed to make that clear in the title or abstract[67]; however, the findings were consistent with our reported results. The second publication did not meet our eligibility criteria because it focused on studies of health economics rather than health outcomes.[68] Interestingly, the study identified similar gaps to the present study, but its scope is outside our protocol and research question. The third was not picked up during screening because the journal is not indexed in the nine predefined data sources for the searches.[69]

The data extraction form may have missed a few potential reporting topics—which will be recovered, in the next stages of the ACCORD project, by additional reviews and the Delphi panel experience. Furthermore, one study was published after our search date, showing that

**Table 2** Studies providing guidance for reporting items in the extraction form of this systematic review

| Reporting items | Studies that provide guidance | |
|---|---|---|
| Background | Number | References |
| 1.1 Rationale for choosing a consensus method over other methods | 4 | 12 25 27 28 |
| 1.2 Clearly defined objective | 6 | 12 17 18 20 27 28 |
| Methods | | |
| 2.1 Review of existing evidence informing consensus study | 5 | 20 21 27 28 31 |
| 2.2 Inclusion and exclusion criteria of the literature search | 3 | 17 20 22 |
| 2.3 Composition of the panel | 16 | 12 16–23 25–30 32 |
| 2.4 Public patient involvement (PPI) | 0 | |
| 2.5 Panel recruitment | 4 | 12 17 22 23 |
| 2.6 Defining consensus and the threshold for achieving consensus | 13 | 12 17–21 23–29 |
| 2.7 Decision of item approval | 3 | 12 17 27 |
| 2.8 Number of voting rounds | 10 | 12 16 18 20 21 23 26–28 32 |
| 2.9 Rationale for number of voting rounds | 8 | 16 20–23 25 26 28 |
| 2.10 Time between voting rounds | 1 | 17 |
| 2.11 Additional methods used alongside consensus | 2 | 17 23 |
| 2.12 Software or tools used for voting | 1 | 25 |
| 2.13 Anonymity of panellists and how this was maintained | 7 | 16 20–22 25 28 29 |
| 2.14 Feedback to panellists at the end of each round | 11 | 17 19–22 25–29 31 |
| 2.15 Synthesis/analysis of responses after voting rounds | 5 | 12 22–24 30 |
| 2.16 Pilot testing of study material/instruments | 3 | 12 22 28 |
| 2.17 Role of the steering committee/chair/co-chair/facilitator | 0 | |
| 2.18 Conflict of interest or funding received | 4 | 12 29 30 32 |
| 2.19 Measures to avoid influence by conflict of interest | 1 | 12 |
| Results | | |
| 3.1 Results of the literature search | 1 | 12 |
| 3.2 Number of studies found as supporting evidence | 0 | |
| 3.3 Response rates per voting round | 5 | 12 21 22 25 30 |
| 3.4 Results shared with respondents | 9 | 12 17 20 25–28 30 31 |
| 3.5 Dropped items | 5 | 12 16 18 26 32 |
| 3.6 Collection, synthesis and comments from panellists | 5 | 12 17 22 28 31 |
| 3.7 Final list of items (eg, for guideline or reporting guideline) | 4 | 12 22 30 31 |
| Discussion | | |
| 4.1 Limitations and strengths of the study | 5 | 12 20 25 27 28 |
| 4.2 Applicability, generalisability, reproducibility | 3 | 12 17 26 |

the development of reporting guidelines for consensus methodologies is an active area, with more studies being published on the topic continuously, which could inform future stages or updates of ACCORD.[70] Comments regarding deficient reporting from the included studies varied from generic statements such as 'reporting could be improved' to rather specific comments of which aspects of consensus methods were inadequately or not reported. Far more detailed data were provided regarding guidance to improve reporting quality or suggestions for items that require reporting. Both composition and characteristics of the panel, and defining consensus and threshold

for achieving assessment received, were consistently addressed and appeared to be critical items that should be reported in sufficient detail. Feedback to the panel might be considered an important aspect of ensuring ongoing engagement with the panellists, transparency and replicability of methods; thus, it was somewhat surprising to see just 11 of the 18 studies consider this an element of consensus methodology worth reporting.

Some items were not addressed in any of the studies, specifically PPI, which is currently considered a key element in the shared decision-making process and is a component of guideline development.[71] Just four

studies made reference to the COI of panel members and project funding. COI of panellists, as well as of chair, cochair and steering committee, can directly or indirectly impact and influence decision-making during the various steps of consensus methodology. As such, COI remains under-reported and is often inconsistently described.[72] This also raises concerns about the measures that can be taken to mitigate the potential influence of COI and to ensure that those panellists who do have relevant interests are, for example, not able to vote on pertinent items. For full transparency and to promote discussion, all data retrieved are reported as supplementary material (online supplemental materials 3–5).

Although conclusions are limited by the paucity of studies, a few were particularly informative. The first was a systematic review on the use and reporting of the Delphi method for selecting healthcare indicators.[17] Specifically, this review not only provided guidance for planning and using the Delphi procedure but also additionally formulated general recommendations for reporting. The second study was a guidance report on consensus methods such as Delphi and NGT, which were used in medical education research.[28] The authors reported that there is a lack of 'standardisation in definitions, methodology and reporting' and proposed items for researchers to consider when using consensus methods to improve methodological rigour as well as the reporting quality. However, it is worth noting that none of these studies followed the EQUATOR Network guidance for the development of a reporting guideline.[3]

The third study we would like to highlight is the Guidance on Conducting and REporting DElphi Studies (CREDES) in palliative care, which was based on a methodological systematic review.[12] This study focused on the development of guidance in palliative care, although it may not be suitable for extrapolation to other biomedical areas. Furthermore, this study only considered the Delphi methodology, whereas we included studies covering consensus processes involving non-Delphi-based methods or 'modified Delphi' in our review (and in the ACCORD project overall). However, many of the suggestions made regarding the design and conduct of Delphi studies in addition to recommendations for reporting are equally applicable to our ACCORD project. These items will be used and integrated into the next step of the project, which is the development of a reporting checklist on consensus methods.

Two additional studies proved to be of particular value.[21 25] One provided a preliminary Delphi checklist to be used for Outcome Measures in Rheumatology.[25] The other concluded, in a scoping review that consensus methods are 'poorly standardised and inconsistently used' and exposed reporting flaws in consensus reports.[21]

## CONCLUSION

The principal objectives of this systematic review were to conduct a comprehensive search and to identify the existing evidence on the quality of reporting of consensus methodology. As such, we have been able to gather together all relevant studies, summarise the existing research and highlight key gaps in the current evidence base on consensus methods. This systematic review will ultimately inform the generation of a draft checklist of items for the development steps of the ACCORD reporting guideline.

**Author affiliations**
[1]Department of Dermatology, Leiden University Medical Center, Leiden, Zuid-Holland, Netherlands
[2]Nuffield Department of Orthopaedics, Rheumatology and Muskuloskeletal Sciences, Centre for Statistics in Medicine, University of Oxford and EQUATOR Network UK Centre, Oxford, UK
[3]Stanford Anesthesia, Informatics and Media Lab, Stanford University School of Medicine, Stanford, California, USA
[4]The BMJ, London, UK
[5]Veritas Health Sciences Consultancy, Huntingdon, UK
[6]Sciwright Limited, Somerset, UK
[7]Global Medical Affairs, Ipsen, Abingdon, Oxfordshire, UK

**Acknowledgements** We would like to thank Jan W. Schoones of the Directorate of Research Policy (formerly: Walaeus Library), Leiden University Medical Centre, Leiden, The Netherlands, for his assistance in the development of the search strategy. Furthermore, we would like to thank the other members of the ACCORD project Steering Committee for their contributions to the protocol: Amrit Pali Hungin, Christopher C Winchester, David Tovey, Keith Goldman, Rob Matheis and Niall Harrison. We are grateful for the editorial guidance and reviewer insights from the BMJ Open team. We agree their observations improved the manuscript.

**Contributors** EJvZ, PL, ZF and WTG contributed to the screening and agreed on the inclusion of studies. EJvZ and WTG extracted data from the included studies. AP, ZF and ELH contributed to the discussion of extracted data and interpretation. EJvZ was the major contributor in the review of studies, data extraction, interpretation of findings as well as writing of the manuscript. All authors read the final manuscript, provided feedback and approved the final manuscript. EJvZ is the guarantor.

**Funding** PL contributed to this study using her time funded by CRUK. WTG contributed his time with the agreement of his employers. Apart from these authors, this study was conducted without external funding for the development of the study design; for the collection, analysis and interpretation of the data; or for the writing of the report. As ZF (Veritas Health Sciences Consultancy) acted as an independent consultant for this work, his support for data extraction and tabulation was funded by Oxford PharmaGenesis. Support for Open Access was also provided by Oxford PharmaGenesis. The decision to submit the paper for publication was made solely by the authors.

**Competing interests** PL is a member of the UK EQUATOR Centre, an organisation that promotes the use of reporting guidelines, many of which are developed using consensus methods, and she is personally involved in the development of other reporting guidelines. ELH has worked with Ogilvy Health UK on consensus projects. WTG is a former employee of Ipsen and is now employed by Bristol Myers Squib. AP is an editor at the BMJ and is a senior research scientist at Stanford University, where she is responsible for advising and seeking funding on Delphi and other research studies. EJvZ and ZF have no conflict of interest.

**Patient and public involvement** Patients and/or the public were involved in the design, or conduct, or reporting, or dissemination plans of this research. Refer to the Methods section for further details.

**Patient consent for publication** Not applicable.

**Ethics approval** Not applicable.

**Provenance and peer review** Not commissioned; externally peer reviewed.

**Data availability statement** All data relevant to the study are included in the article or uploaded as supplementary information. All key data for this study are included in this article or uploaded as online supplementary information. The ACCORD protocol has been listed on the EQUATOR website (Reporting guidelines under development for other study designs | The EQUATOR Network (equator-

network.org)) and registered with the Open Science Framework (https://osf.io/2rzm9).

**ORCID iD**
Esther J van Zuuren http://orcid.org/0000-0002-4780-0182

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
