## [Reviewer comments · BMJ Open]

ARTICLE DETAILS

TITLE (PROVISIONAL)	Existing guidance on reporting of consensus methodology: a systematic review to inform ACCORD guideline development
AUTHORS	van Zuuren, Esther; Logullo, Patricia; Price, Amy; Fedorowicz, Zbys; Hughes, Ellen L; Gattrell, William T

VERSION 1 – REVIEW

REVIEWER	Rodrigues, Myanca Hamilton
REVIEW RETURNED	28-Jun-2022

GENERAL COMMENTS	Great paper - no comments.
----------------------------

REVIEWER	Grant, Sean Indiana University Richard M Fairbanks School of Public Health
REVIEW RETURNED	11-Jul-2022

GENERAL COMMENTS	I have been asked to review “Existing guidance on reporting of consensus methodology: a systematic review to inform ACCORD guideline development” for consideration at BMJ Open. I enjoyed reading the manuscript and am optimistic that the ACCORD Guideline will eventually have a positive impact on research using consensus methods. I offer the following comments (in order as they appear in the manuscript) to consider in a revision. Major comments: 1. Methods—Data Extraction: Please explain how the authors identified and created the potential checklist items in the data extraction form. This form and its organization has the potential to significantly influence what the data extractors attended to, what data were collected, and thus the eventual ACCORD checklist.2. Methods/Results: Related to the above, I am surprised not to see items on the following in the potential checklist data extraction tool and/or the findings: a meeting of participants, stopping criteria, added items, interpretation of results (discussion)• Results: The authors do not provide data for the objective of the study (i.e., “to identify evidence on the quality of reporting of consensus methodology”). Aside from one sentence (“The majority of studies indicated that reporting of consensus methods could be improved and summarised current limitations in reporting or proposed suggestions for improvement”), the results only summarize potential checklist items from their search. Either please add data on actual reporting quality, or clarify in the objectives that the purpose of this study was solely to identify potential items from existing guidance.
---

- Results—Data Extraction. Please check for errors in reported information. For example, the text says there are 19 items in the data extraction form, whereas the copy provided with the manuscript has 30 items. Errors in the denominator can impede reader confidence in reported results.
- Results: The authors are missing studies that I believe are eligible for their review (e.g., <https://journals.plos.org/plosmedicine/article?id=10.1371/journal.pmed.1000393>, <https://link.springer.com/article/10.1007/s40273-016-0425-9>, <http://www.openscienceonline.com/journal/archive2?journalId=718&paperId=3586>, [https://linkinghub.elsevier.com/retrieve/pii/S1865-9217\(22\)00076-9](https://linkinghub.elsevier.com/retrieve/pii/S1865-9217(22)00076-9)). The Sinha article is a particular omission, especially as it is cited in included studies, calling into question reader confidence in this statement: “Checking of the references of these full-text publications did not yield any additional eligible articles). Please check with the information scientist as to why these studies were missed, and update the search accordingly.
- Results: Please provide information on the degree to which include studies focused on Delphi, NGT, consensus conferences, or other methods. From my read, included studies and items seem rather specific to Delphi, which could have implications for ACCORD (e.g., under-coverage or inapplicability of items to other consensus methods).
- Results: I recommend adding another narrative section and table in the main text (rather than just the supplement) that summarizes the actual checklist items in included studies. The text right now only provides counts of studies with items in the potential checklist (i.e, Table 1); as an active researcher in this area, I am interested in knowing specifically what these included studies recommended in each of these high-level sections.
- Discussion: The Discussion section reads like a results section, providing more information on specific findings not found in the actual Results section. Please revise the Discussion to summarize what was found, implications for the field, and limitations of this study.

Minor comments:

- Methods: The authors write that the review conforms to PRISMA, though PRISMA is a guideline for reporting, not conduct. Please clarify that this *manuscript* conforms to PRISMA, and then please cite any guidance used to *design and conduct* the review (if any was used).
- Methods: OSF stands for Open Science *Framework*, not *Foundation*.
- Methods—Inclusion Criteria: please provide more concrete details on how the authors operationalized “reviews” and “published guidance”. For example, did the reviews have to be a systematic review? And what distinguishes “guidance” from an “editorial”?
- Methods—Literature Search Strategy: I am confused by this sentence “We chose to keep the search terms broad (in not using the singular terms "delphi" and "consensus" but always in phrases or with other contextual words).” To me, not using the singular term “Delphi” makes the search more narrow, not more broad. Could the authors please clarify what they mean here?
- Table 1—To me, the “results of the literature search” should not be placed in the “Results” of a checklist for the Delphi method. While a literature review may inform a Delphi study, the review itself is not the result of the Delphi, but a study in its own right. As such, should this not be placed in the “Methods” of the Delphi checklist, perhaps combined with Item 2.2 as one item on literature informing the Delphi?
- Results—Data Extraction: The authors combine “composition of the panel” and “criteria for selecting the panelists” as one item. I disagree with this combination, as most reporting guidelines (e.g., PRISMA, CONSORT) separate eligibility criteria (“criteria for selecting the panelists”) from baseline data (“composition of the panel”). I recommend separating these and moving “composition of the panel” to the results (per other reporting guidelines).
- Results—Data Extraction: Please report actual figures and please be precise in reporting results. For example, “the prespecified number of voting rounds and

provision of feedback to the panellists at the end of each round was addressed in half of the studies.” However, the former is reporting in 10 studies, the latter in 11 studies, and “half” of the studies is 9, not 10 or 11.

VERSION 1 – AUTHOR RESPONSE

Reviewer: 1

Ms. Myanca Rodrigues, Hamilton

Comments to the Author:

Great paper - no comments.

RESPONSE R1. We thank R1 for the compliment and approval of the paper.

Reviewer: 2

Dr. Sean Grant, Indiana University Richard M Fairbanks School of Public Health Comments to the Author:

I have been asked to review “Existing guidance on reporting of consensus methodology: a systematic review to inform ACCORD guideline development” for consideration at BMJ Open. I enjoyed reading the manuscript and am optimistic that the ACCORD Guideline will eventually have a positive impact on research using consensus methods.

I offer the following comments (in order as they appear in the manuscript) to consider in a revision.

RESPONSE: We are grateful for your detailed review, and we addressed each of the issues raised below. We agree with you that the ACCORD guideline will have an impact on how research using consensus methods is reported in the literature. We thank you for your contribution to this project.

Major comments:

R2-1. Methods—Data Extraction: Please explain how the authors identified and created the potential checklist items in the data extraction form. This form and its organization has the potential to significantly influence what the data extractors attended to, what data were collected, and thus the eventual ACCORD checklist.

RESPONSE: The potential topics used in our extraction form were not exactly (not just yet) “checklist items”, but rather “reporting issues”. These were inspired by: a) the authors’ own experience with reporting quality evaluation; and b) the reading of the literature on consensus methods (covering studies not included in the review) when writing up our study protocol, which is now published. The extraction form contained 2 fields for “additional topics” that a) the extractors felt were not directly addressed by the included studies but were rather inferences about topics that could be potential issues in consensus reporting and b) topics addressed by the included studies that were not captured through the data extraction form and which we felt could be useful.

It is important to highlight, though, that the items that arose from this systematic review are not the only items that will be covered by the ACCORD reporting guideline we are developing. We have reworded the text under data extraction to clarify this accordingly. The development of the reporting guideline will cover other steps recommended by the EQUATOR Network, such as surveys (Delphi or not) and consensus meetings. Final refinement or wordsmithing of the checklist items, including the perspectives of lay people, will guarantee the items are clear and free from ambiguity. This process

as a whole, we expect, will minimize the risk of bias from the systematic review extraction or the extractors' points of view.

As there is no standardised extraction form for systematic reviews for reporting guidelines (that are not based on a clear-cut clinical question, please see <https://doi.org/10.1186/s12874-020-01143-3>), we did our best, according to general published recommendations, to provide extractors with instructions about what to collect from the papers, and collect text excerpts whenever possible, testing the extraction to anticipate conflicts, perform double extraction and conflict resolution meeting (using Covidence). We are, therefore, confident that the content extracted from the papers reviewed is consistent.

We still hope that this peer reviewer accepts the invitation sent by the Steering Committee to participate in the Delphi panel as it would be a privilege to have Dr Grant on board with his expertise.

R2-2.Methods/Results: Related to the above, I am surprised not to see items on the following in the potential checklist data extraction tool and/or the findings: a meeting of participants, stopping criteria, added items, interpretation of results (discussion)

RESPONSE: We completely acknowledge the reviewer's concern, which makes sense. Although we used a sensitive search, we found only 18 publications that matched our inclusion criteria. From these, 4 were about Delphi plus other consensus methods and the remaining 14 were heavily based on the Delphi method only. This is one of the reasons why, for example, "meetings" was not a very frequent topic in the studies reviewed because currently Delphi studies are conducted online — but it is something we must, and we will address in the next steps of the reporting guideline development. Stopping criteria, added items and interpretation of results were not anticipated in our extraction form, but showed as possible additional items in this review and in the broader literature we are reviewing about methodology. So, these items will be submitted to the ACCORD Delphi panel and discussed for inclusion.

R2-3 - Results: The authors do not provide data for the objective of the study (i.e., "to identify evidence on the quality of reporting of consensus methodology"). Aside from one sentence ("The majority of studies indicated that reporting of consensus methods could be improved and summarised current limitations in reporting or proposed suggestions for improvement"), the results only summarize potential checklist items from their search. Either please add data on actual reporting quality, or clarify in the objectives that the purpose of this study was solely to identify potential items from existing guidance.

RESPONSE: This is a very good point, and we thank the reviewer for this observation. We have added an additional section at the start of Results as well as an additional Table. The papers reviewed did not provide a systematic or standardised "quality of reporting evaluation", as expected, since there is no standardised reporting guidance available on which they could be based. The studies included evaluated the literature critically and offered insights into the gaps of information about consensus. Some of them went forward to make recommendations — but this was done based solely on the authors' opinion, not on a systematic approach to reporting guidance development. We added this comment to our Results section, and we hope this clarifies the issue. Furthermore, we added a new supplemental table with data about reporting quality of the included studies. We also address this in our Discussion.

R2-4 - Results—Data Extraction. Please check for errors in reported information. For example, the text says there are 19 items in the data extraction form, whereas the copy provided with the manuscript has 30 items. Errors in the denominator can impede reader confidence in reported results.

RESPONSE: Thank you for such a careful and thorough review of our paper. However, we are not sure there is such a discrepancy. We could not find where the text refers to 19 items in the data extraction form in the paper. What we did state was that three studies addressed between 12 and 19 items in the data extraction form — not that our extraction form had only 19 items. Maybe it was a misunderstanding? To avoid such ambiguity, we have rephrased this now as “three studies addressed between 12 and 19 reporting items of the 30 items present in the data extraction form of this review”.

R2-5 - Results: The authors are missing studies that I believe are eligible for their review (e.g., <https://journals.plos.org/plosmedicine/article?id=10.1371/journal.pmed.1000393>, <https://link.springer.com/article/10.1007/s40273-016-0425-9>, <http://www.openscienceonline.com/journal/archive2?journalId=718&paperId=3586>, [https://linkinghub.elsevier.com/retrieve/pii/S1865-9217\(22\)00076-9](https://linkinghub.elsevier.com/retrieve/pii/S1865-9217(22)00076-9)). The Sinha article is a particular omission, especially as it is cited in included studies, calling into question reader confidence in this statement: “Checking of the references of these full-text publications did not yield any additional eligible articles). Please check with the information scientist as to why these studies were missed, and update the search accordingly.

RESPONSE: We thank the reviewer for suggesting additional papers we could have missed. However, the first three studies cited are not about the quality of reporting in Delphi or in consensus exercises; they are studies using Delphi to select outcomes for clinical research. Core outcome selection, as intensively studied by the James Lind Alliance (COMET), is indeed a major field for using Delphi. That does not mean these studies are about the quality of reporting of Delphi, though. The second study cited discusses reporting criteria for two types of study design to use expert judgement in model-based Economic Evaluations, that is not biomedical research. The fourth study is indeed very interesting but not published within the scope of our search dates (up to 7 January 2022). But we will consider it for the next steps of our project. Thank you for highlighting this study.

R2-6 - Results: Please provide information on the degree to which include studies focused on Delphi, NGT, consensus conferences, or other methods. From my read, included studies and items seem rather specific to Delphi, which could have implications for ACCORD (e.g., under-coverage or inapplicability of items to other consensus methods).

RESPONSE: As highlighted above, yes, the reviewer is right in saying that the included studies focused mostly on Delphi. This is now added under characteristics of included studies, as well as in the Discussion. It is indeed unfortunate, as we aim to address other types of consensus methods used in isolation or in conjunction with Delphi in health research. However, we cannot determine what is published in the literature; we can only expect that more studies will be conducted in the future addressing the quality of reporting in methods like the nominal group technique (NGT) and the widely used — and poorly described — consensus meetings with experts. During the process of development of the ACCORD guideline, we are, however, looking at studies that did not aim to evaluate reporting but did discuss these non-Delphi methodologies. These are papers about “how to conduct”, “steps to take”, and “advantages and disadvantages” of other methodologies. Again: although not focused on reporting, they do offer insight on issues that are important when one tries to interpret the results of studies using these techniques. Therefore, anything that may incur bias is interesting from the perspective of reporting quality and will, hopefully, be addressed by the ACCORD Statement and ACCORD Explanation & Elaboration documents to be published. We just want to remind the reviewer

that the paper under evaluation here is just the initial systematic review of the literature — not an article about the results of whole process of developing ACCORD.

R2-7 - Results: I recommend adding another narrative section and table in the main text (rather than just the supplement) that summarizes the actual checklist items in included studies. The text right now only provides counts of studies with items in the potential checklist (i.e, Table 1); as an active researcher in this area, I am interested in knowing specifically what these included studies recommended in each of these high-level sections.

RESPONSE: We thank the reviewer for the suggestion, but we beg to disagree. This was not our research question and would therefore be considered selective reporting. As we commented above, the few “checklists” on “what to report” provided by some, but not all included papers were not systematically developed. They were based on the (valuable) opinion of the authors. By reproducing such items, we could give the impression that we are endorsing them. And the development of a reporting guideline involves several steps of discussion, selection, consensus and wordsmithing that will result, eventually, in a checklist that is appropriate to use. In summary, we don’t want to summarize the checklists’ items from these studies, not just yet, and we hope that the editor understands our approach.

R2-8 - Discussion: The Discussion section reads like a results section, providing more information on specific findings not found in the actual Results section. Please revise the Discussion to summarize what was found, implications for the field, and limitations of this study.

RESPONSE: We can understand why the reviewer feels like the Discussion was repeating Results. We added more information to the Results section to try and address this. However, in the Discussion section we have more freedom to interpret and offer our views about the studies included — something which is not appropriate for inclusion in the Results section. So, all studies cited in the Discussion are present and cited in the Results section first, we utilized the Discussion section to acknowledge their positive aspects or critic the weaker points in the Discussion. We have summarized the Discussion results and we have reordered our Discussion to accommodate the reviewer’s guidance.

Minor comments:

R2-9 - Methods: The authors write that the review conforms to PRISMA, though PRISMA is a guideline for reporting, not conduct. Please clarify that this *manuscript* conforms to PRISMA, and then please cite any guidance used to *design and conduct* the review (if any was used).

RESPONSE: The reviewer is absolutely right. PRISMA is guidance for reporting, not methods or conduction of studies. We have added the word “manuscript” to clarify this. We thank the reviewer for spotting this.

R2-10 - Methods: OSF stands for Open Science *Framework*, not *Foundation*.

RESPONSE: Thank you very much for picking this up, we wrote it once correctly and once incorrectly and now both are listed correctly.

R2-11 - Methods—Inclusion Criteria: please provide more concrete details on how the authors operationalized “reviews” and “published guidance”. For example, did the reviews have to be a systematic review? And what distinguishes “guidance” from an “editorial”?

RESPONSE: Thank you for this question. We chose to keep the inclusion criteria broad in such a way that both narrative reviews, as well as methodological reviews or systematic reviews, could be included. The same holds true for ‘published guidance’: we kept this broad as we did not want to exclude valuable articles. With the pilot searches, we noticed that by framing the search terms too tightly, important articles would not be retrieved. Editorials in general express the views of the journal’s editor or of one author about another article published in the same journal. Although these pieces might give some broad insight, they are usually not reporting on a structured evaluation or providing detailed and extensive information. This is why we decided to exclude them. We described the studies better in the manuscript.

R2-12 - Methods—Literature Search Strategy: I am confused by this sentence “We chose to keep the search terms broad (in not using the singular terms “delphi” and “consensus” but always in phrases or with other contextual words).” To me, not using the singular term “Delphi” makes the search more narrow, not more broad. Could the authors please clarify what they mean here?

RESPONSE: The reviewer is correct. We have asked the librarian to rephrase this. It now reads as “We chose to keep the search terms specific (in not using the singular terms “delphi” and “consensus” and included these words with relevant phrases or with other contextual words). In this way, the refined search strategy was better aligned with our inclusion criteria and the objectives of the systematic review.”

R2-13 - Table 1—To me, the “results of the literature search” should not be placed in the “Results” of a checklist for the Delphi method. While a literature review may inform a Delphi study, the review itself is not the result of the Delphi, but a study in its own right. As such, should this not be placed in the “Methods” of the Delphi checklist, perhaps combined with Item 2.2 as one item on literature informing the Delphi?

RESPONSE:

This is quite an interesting question, and we will certainly consider this in the development of the ACCORD reporting guideline when we consult with the Steering Committee and the Delphi panel. Sometimes, a literature review (systematic or not) is used to support consensus exercises, for example, a Delphi survey or a consensus meeting or even the voting in an NGT meeting. Sometimes, the consensus exercises (whether a survey, a meeting or any other) are not based on a previous literature review, either because no evidence is available (the whole point, sometimes, for using consensus) or because the available literature is poor or even because the developers want to obtain the participants’ views “ uncontaminated” by the existing literature (for example, a consensus exercise on the core outcomes to study according to patients’ perspectives). So, to begin with, the “results of the literature search” might or might not be present in the report of consensus. Another thing to consider is whether one or more manuscripts are published about the health issue: there might be a first paper reporting on the results of a literature review — and here, the “Results of the literature search” item belongs to Results — and another for the consensus exercise results, and there might be situations in which both, literature review and consensus are reported in the same paper — in these cases, the authors might consider that the results of the literature review based the main consensus exercise, therefore they might use the results of review just as a base and report them in Methods (“we presented these literature results to the participants”). For now, we will keep

this in the “Results” section. We realized the title of Table 1 is misleading, we have changed this into “Studies providing guidance for reporting items in the extraction form of this systematic review”.

R2-14 - Results—Data Extraction: The authors combine “composition of the panel” and “criteria for selecting the panelists” as one item. I disagree with this combination, as most reporting guidelines (e.g., PRISMA, CONSORT) separate eligibility criteria (“criteria for selecting the panelists”) from baseline data (“composition of the panel”). I recommend separating these and moving “composition of the panel” to the results (per other reporting guidelines).

RESPONSE: We don’t fully understand this comment as “selection of composition of panel” are not part of the PRISMA nor the CONSORT checklists. What we mean by composition of the panel is that it should be described clearly and in full: who will be invited (e.g., clinicians, methodologists, health policy makers, patients), what was their gender, age, region or country of origin, ethnicity. We also think that these choices should be justified. This content item might well be part of the Results section of a manuscript, as what was planned does not always reflects what really happened. But it is a methodological choice in the first instance, and as such is described in our Methods first.

R2-15 - Results—Data Extraction: Please report actual figures and please be precise in reporting results. For example, “the prespecified number of voting rounds and provision of feedback to the panelists at the end of each round was addressed in half of the studies.” However, the former is reporting in 10 studies, the latter in 11 studies, and “half” of the studies is 9, not 10 or 11.

RESPONSE: The reviewer is correct and we have now added the actual numbers.

*** **

COI statements:

Reviewer: 1

Competing interests of Reviewer: None.

Reviewer: 2

Competing interests of Reviewer: I am a research team member for ExpertLens (an online platform and methodology for conducting modified-Delphi studies). My spouse is a salaried employee of, and owns stock in, Eli Lilly and Company.

VERSION 2 – REVIEW

REVIEWER	Grant, Sean Indiana University Richard M Fairbanks School of Public Health
REVIEW RETURNED	27-Jul-2022

GENERAL COMMENT S	I thank the authors for their thoughtful reply and revisions to the originally submitted manuscript. I agree with the authors that the changes make the manuscript more clear and include important additional details, which I found both helpful and interesting as a reader. I offer the remaining comments/questions for the editor and authors to consider:
--

- I appreciate the details provided by the authors in their response letter (R2-1) on how they created the data extraction form, though I do not see this explanation in the paper itself (apologies if I've missed it). Please include in the manuscript how the authors chose the potential topics/reporting issues to include as items in the data extraction form.

- Related to the above, I am not sure the response to R2-2 fully addresses my previous comment (apologies for being unclear). While I understand that few publications addressed these items, I was also surprised that the *data extraction tool* itself did not contain items on a meeting of participants, stopping criteria, added items, and interpretation of results (discussion). This omission led to my curiosity as to how the data extraction tool was developed. As such, in addition to adding the above-requested details to the Methods section, I recommend including a statement in the Discussion that the data extraction form may be missing potential topics/reporting issues.

- Related to the above, the paper is still missing a dedicated "Limitations" section.

- I respectfully disagree with the author response to R2-5. The first study (<https://journals.plos.org/plosmedicine/article?id=10.1371/journal.pmed.1000393>) is a systematic review of Delphi studies in the core outcome set field, and it explicitly has reviewing the quality of *reporting Delphi studies* as an aim (e.g., it states in the "Main Points" that it identified variability in the quality of reporting Delphi studies, Table 1 is called "Reporting quality of the 15 included studies", and Table 2 is a checklist of recommended reporting items for Delphi studies). The second study (<https://link.springer.com/article/10.1007/s40273-016-0425-9>) is about model-based economic evaluations (EEs) *in healthcare*, was published in a *Health* Economics journal (PharmacoEconomics), and involves authors in academic departments of *medicine, health economics, and public health*. The authors do not provide a reason for excluding the third study (<http://www.openscienceonline.com/journal/archive2?journalId=718&paperId=3586>) . I understand the fourth study was published after the search was conducted— perhaps it can be mentioned in the Discussion as evidence that this is an active area with more studies being published on the topic that could inform future stages of ACCORD?

- I am not sure I understand the author response to R2-7? The authors already report the requested information in supplementary files for this paper (which indicates that this information is relevant to the research objectives/questions). I am suggesting that the authors synthesize/summarize this information in the narrative as well so that the reader knows what to take away from these supplements. Perhaps I am misunderstanding what the authors mean by "selective reporting", which IME refers to results deliberately not being fully or accurately reported in order to suppress negative or undesirable findings?

- My previous comment and the author response to R2-8 led me to realize that this article doesn't have a "data analysis" section in the manuscript, and this may be leading to some confusion. Per the second objective of the study ("to select potential checklist items for the ACCORD project to develop a consensus reporting guideline"), I was expecting to see results from a qualitative analysis of included studies, though I am starting to infer that the authors meant to purely describe quantitative descriptive data as the results. Could this section please be added?

- Regarding the author response to R2-13, perhaps a better item name would be "whether the study included a literature review"? I still think the full *results* of a literature review should be its own paper/output, though appreciate the importance of indicating whether such a review was done and what it entailed would be important to summarize in a Delphi study paper (which is why I personally would

	put this item in the “Methods” section of a checklist). If nothing else, food for thought for future stages of ACCORD! - Apologies if I was unclear in R2-14. My point is that reporting guidelines typically have one item in the “Methods” section of the checklist related to study eligibility criteria (e.g., participants to include in a trial) and then a related but distinct item in the “Results” section on the actual characteristics of the sample (e.g., actual demographics for recruited participants). I see these as similar to an item in the “Methods” section of the ACCORD checklist on “criteria for selecting the panelists” and a related but distinct item in the “Results” section on the actual “composition of the panel”. Using an example from a recent Delphi panel of mine (https://www.sciencedirect.com/science/article/abs/pii/S0955395921002887), our “criteria for selecting the panelists” were membership in one of several stakeholder groups (advocates, persons with lived experience, healthcare providers, human/social service practitioners, policymakers, and researchers), while Table 2 summarizes the actual “composition of the panel”.
--	--

VERSION 2 – AUTHOR RESPONSE

Reviewer: 2

Dr. Sean Grant, Indiana University Richard M Fairbanks School of Public Health Comments to the Author:

0. I thank the authors for their thoughtful reply and revisions to the originally submitted manuscript. I agree with the authors that the changes make the manuscript more clear and include important additional details, which I found both helpful and interesting as a reader.

0 RESPONSE: We are glad to know that the reviewer found the manuscript easier to understand now, and interesting as a reader. This is important feedback to the Editor.

I offer the remaining comments/questions for the editor and authors to consider:

@R2-1 I appreciate the details provided by the authors in their response letter (R2-1) on how they created the data extraction form, though I do not see this explanation in the paper itself (apologies if I’ve missed it). Please include in the manuscript how the authors chose the potential topics/reporting issues to include as items in the data extraction form.

R2-1 RESPONSE: Thank you for this comment; we agree with the reviewer that this information is important for the journal reader. We have now added the following sentence to the manuscript, as suggested, under Data extraction.

“The data extraction form questions were compiled based on the review authors’ own experiences with reporting quality evaluation and literature on consensus methodology. Furthermore, two additional free text fields were created for extractors to present issues addressed by the included studies that were not captured by the other questions, and for others that the extractors felt were not

directly addressed by the studies but were rather inferences about topics that could be potential issues in the reporting of consensus methods.”

@R2-2 Related to the above, I am not sure the response to R2-2 fully addresses my previous comment (apologies for being unclear). While I understand that few publications addressed these items, I was also surprised that the *data extraction tool* itself did not contain items on a meeting of participants, stopping criteria, added items, and interpretation of results (discussion). This omission led to my curiosity as to how the data extraction tool was developed. As such, in addition to adding the above-requested details to the Methods section, I recommend including a statement in the Discussion that the data extraction form may be missing potential topics/reporting issues.

R2-2 RESPONSE: We have added some sentences to the Discussion to acknowledge that the data extraction form may have missed potential reporting topics. We also took care to add this under the “STRENGTHS AND LIMITATIONS OF THIS STUDY” section on the first page. However, issues like meetings, stopping criteria and others, noted by the reviewer, although not present in our extraction form closed questions, were raised by the extractors in its free-text field and did come up as topics of interest, albeit using different terminology.

- Related to the above, the paper is still missing a dedicated “Limitations” section.

R2-2-b RESPONSE: We opted not to build a separate section of strengths and limitations within the manuscript for two reasons: first, limitations and strengths of the study are intertwined in the Discussion text as a whole (one example is the first paragraph); and second, the journal provides a separate section about the strengths and limitations after the Abstract, to be published as a summary box in the final typeset article. So not only they are discussed in the article, they are highlighted in this separate box.

@R2-5 I respectfully disagree with the author response to R2-5. R2-5a The first study (<https://journals.plos.org/plosmedicine/article?id=10.1371/journal.pmed.1000393>) is a systematic review of Delphi studies in the core outcome set field, and it explicitly has reviewing the quality of *reporting Delphi studies* as an aim (e.g., it states in the “Main Points” that it identified variability in the quality of reporting Delphi studies, Table 1 is called “Reporting quality of the 15 included studies”, and Table 2 is a checklist of recommended reporting items for Delphi studies).

@R2-5a RESPONSE: As the reviewer points out, it is indeed “a systematic review of Delphi studies”. But it is not as explicit to say it is a review on reporting quality — or at least it did not have this laid out as an objective.

As we reported in our Methods section, the screening, in our study, was based on title and abstract, as common practice in systematic reviews. Our pre-set eligibility criteria was that, to be included, a study had to be focused on reporting quality of studies on consensus methodology in health.

The title of the article by Sinha and colleagues is: “*Using the Delphi technique to Determine Which Outcomes to Measure in Clinical Trials: Recommendations for the Future Based on a Systematic Review of Existing Studies*”. No clue here about reporting quality.

The study abstract (a quite short one) is: “*Ian Sinha and colleagues advise that when using the Delphi process to develop core outcome sets for clinical trials, patients and clinicians be involved, researchers and facilitators avoid imposing their views on participants, and attrition of participants be minimized.*”

Both title and abstract suggest the review is about what methods should be used in studies using Delphi for core outcomes selection (COS) for clinical trials. They do not suggest that the paper would discuss the reporting quality of consensus methods, but rather, consensus methodologies to take into account during COS development. So we did not retrieve the full paper — although it is interesting reading about consensus methodology, it did not fulfil the inclusion criteria of this particular systematic review.

When one opens the full text, the study objectives declares: “methodological considerations or reporting for studies using the Delphi technique to determine which outcomes or domains to measure in clinical research studies” — which also seems to be focusing on methodology of Delphi for COS, not on the reporting quality of the papers.

In hindsight, this paper did include some useful elements about how to conduct studies and a few comments on reporting, as the authors found out that reporting was poor. This finding, though, was incidental and not the objective of their review. Also, the comments about reporting could not have been foreseen based on title or abstract during screening. Therefore, we reported, completely, rigorously and transparently, what we did, how, when, and what we found in our review and, according to the review protocol, we were not wrong in not including this study in the initial screening of this review.

We must remind the reviewer that the all the topics inspired by this systematic review will be included in the Delphi survey for the ACCORD development, as described in our protocol; however, these are not the ‘only’ items. The systematic review being reported here is not the only base of the ACCORD project development. The Sinha et al. paper and its insights will be considered by the Steering Committee, along with over 40 other papers, to draft the initial list of the ACCORD reporting guideline to be presented to the Delphi panel — for which, again, we invite the reviewer to be part of. The panel will be able to suggest new items based on this and any other paper.

@R2-5b The second study (<https://link.springer.com/article/10.1007/s40273-016-0425-9>) is about model-based economic evaluations (EEs) *in healthcare*, was published in a *Health* Economics journal (PharmacoEconomics), and involves authors in academic departments of *medicine, health economics, and public health*.

@R2-5b RESPONSE: The inclusion criteria in our systematic review, as stated in the manuscript, was: “*Eligible studies consisted of reviews and published guidance which addressed the reporting*

quality of consensus methodology and aimed to improve health outcomes in biomedicine or clinical practice.”

As explained in the response above and in the paper, the screening was based on title and abstract, using the Rayyan platform. The title “*Reporting Guidelines for the Use of Expert Judgement in Model-Based Economic Evaluations*” did not suggest directly that it aimed to improve health outcomes in biomedicine or clinical practice (nor did the abstract). Therefore, it was not included during the screening process.

The reviewer is correct in saying that the paper by Iglesias et al. contains useful elements for the ACCORD reporting guideline, and by reading the full text we can see a rich discussion about the use of the terms “expert opinion” versus “expert elicitation”. The paper will be considered at a later stage of ACCORD development, albeit not included in this systematic review.

@R2-5c The authors do not provide a reason for excluding the third study (<http://www.openscienceonline.com/journal/archive2?journalId=718&paperId=3586>).

R2-5c RESPONSE: We did not exclude this study: it was simply not retrieved by our search. The journal where it was published was not indexed in any of the databases we searched:

- Web of Science - core collection
- MEDLINE (Web of Science)
- PubMed
- MEDLINE (OVID)
- Embase (OVID)
- Cochrane Library
- Emcare (OVID)
- Academic Search Premier
- PsycINFO

The article can only be found via Google Scholar using this query "Delphi"|"nominal group"|"nominal grouping"|"nominal groups" "Reporting Guidelines"|"Reporting Guideline"|"Reporting Quality"|"Quality of Reporting"

This manuscript is reporting the methods and results of this specific systematic review, and we cannot include it retrospectively now. However, as stated above for the other studies that were not included, we thank the reviewer for having pointed them out and we will examine these papers for the ACCORD project development.

@R2-5d I understand the fourth study was published after the search was conducted—perhaps it can be mentioned in the Discussion as evidence that this is an active area with more studies being published on the topic that could inform future stages of ACCORD?

@R2-5d RESPONSE: We welcome the suggestion by the reviewer about mentioning this paper, published online on June 2022 (ahead of print). We have added this text to the manuscript:

“... one study was published after our search date, showing that the development of reporting guidelines for consensus methodologies is an active area, with more studies being published on the topic continuously, which could inform future stages or updates of ACCORD.[69]”

@R2-7 I am not sure I understand the author response to R2-7? The authors already report the requested information in supplementary files for this paper (which indicates that this information is relevant to the research objectives/questions). I am suggesting that the authors synthesize/summarize this information in the narrative as well so that the reader knows what to take away from these supplements. Perhaps I am misunderstanding what the authors mean by “selective reporting”, which IME refers to results deliberately not being fully or accurately reported in order to suppress negative or undesirable findings?

@R2-7 RESPONSE: We had understood that the reviewer’s request was for us to provide the checklist items as listed in the included studies. The reviewer now indicates that what he meant was content that is already in the supplementary files.

What we meant with “selective reporting” was adding data in Result section which did not address our research question, i.e. going beyond what we agreed on in our protocol. We hope this clarifies the issue and misunderstandings.

The reviewer suggests we synthesise/summarise the findings — or the reporting issues raised by the papers included. As we said before, we will not reproduce the checklists in the supplementary files, we have only indicated when papers includes such a list in online Supplementary Material file 4 and 5. These checklists were based solely on the authors’ opinions, not on a systematic approach to reporting guidance development.

The synthesis that the reviewer requests is provided under the heading “Data extraction” in the Results section and in Table 1, where we summarise the “items that are not or are not adequately reported in sufficient detail”. Therefore, we believe we solved the problem and provided the reader with a clear summary of reporting issues on consensus studies as evaluated by the papers included.

@R2-8 My previous comment and the author response to R2-8 led me to realize that this article doesn’t have a “data analysis” section in the manuscript, and this may be leading to some confusion. Per the second objective of the study (“to select potential checklist items for the ACCORD project to develop a consensus reporting guideline”), I was expecting to see results from a qualitative analysis of included studies, though I am starting to infer that the authors meant to purely describe quantitative descriptive data as the results. Could this section please be added?

@R2-8 RESPONSE: The reviewer is correct: we did not plan to do any analysis. As we state in the abstract (Data extraction and synthesis) “*Reporting quality items addressed by the studies were synthesized narratively.*” We have summarized all data narratively in the Results section as well as in

the Supplementary Material. And again, the reviewer is correct: we did not have a “data analysis” section in our manuscript, which we added now, clarifying:

“The topics extracted and the methods used in the studies included are synthesised narratively, in text and tables (and Supplementary Material). No further analyses were carried out but these will follow during the next stage of the ACCORD project as per protocol.”

We followed the stages recommended by the EQUATOR Network, and our own study protocol, and we prefer to be as transparent as possible about this second stage or our protocol.

A draft checklist is provided in Online Supplementary Material 6. As the reviewer probably noticed in our second round of review, we have not adjusted the original text from the original studies, and opted to cite them in quotation marks exactly as reported. This is because we feel it is clear enough and in concordance with our protocol, i.e., it is not yet the time to edit and refine the checklist, this will be done by our Steering Committee in conjunction with the ACCORD Delphi panel.

@R2-13 Regarding the author response to R2-13, perhaps a better item name would be “whether the study included a literature review”? I still think the full *results* of a literature review should be its own paper/output, though appreciate the importance of indicating whether such a review was done and what it entailed would be important to summarize in a Delphi study paper (which is why I personally would put this item in the “Methods” section of a checklist). If nothing else, food for thought for future stages of ACCORD!

@R2-13 RESPONSE: We thank the reviewer for this comment and we already discussed this within the Steering Committee. Indeed, food for thought. This issue will be addressed before the first Delphi round.

@2-14 Apologies if I was unclear in R2-14. My point is that reporting guidelines typically have one item in the “Methods” section of the checklist related to study eligibility criteria (e.g., participants to include in a trial) and then a related but distinct item in the “Results” section on the actual characteristics of the sample (e.g., actual demographics for recruited participants). I see these as similar to an item in the “Methods” section of the ACCORD checklist on “criteria for selecting the panelists” and a related but distinct item in the “Results” section on the actual “composition of the panel”. Using an example from a recent Delphi panel of mine (<https://www.sciencedirect.com/science/article/abs/pii/S0955395921002887>), our “criteria for selecting the panelists” were membership in one of several stakeholder groups (advocates, persons with lived experience, healthcare providers, human/social service practitioners, policymakers, and researchers), while Table 2 summarizes the actual “composition of the panel”.

@2-14 RESPONSE: We thank the reviewer for clarifying and we understand the point now. This was indeed not considered in our data extraction form, but this needs to be addressed during further development of the checklist as correctly pointed out.

VERSION 3 – REVIEW

REVIEWER	Grant, Sean Indiana University Richard M Fairbanks School of Public Health
REVIEW RETURNED	12-Aug-2022

GENERAL COMMENTS	I thank the authors for their attentiveness to reviewer comments in this revision to the paper. I found most responses (e.g., including more information on the data extraction form) to be helpful and improve the manuscript. I offer some remaining thoughts on the revised paper for the authors and editors to consider. Aside: As context for the below, I would like to clarify that I trust in specific future stages of ACCORD and the project overall to include the results reported in this manuscript and other findings of the project. That said, I do still think it important to critically appraise this review as a stand-alone piece, and thus that limitations should be addressed or stated explicitly in this paper (even if these limitations are planned to be addressed in future stages of the project). And after this second revision, I believe that there are some methodological issues where the authors and I respectfully though fundamentally disagree, in which case the editor will need to make a decision. Namely, one underlying issue is that the authors and I appear to disagree on the permissibility of retrospectively adding analyses. In general, I think it allowable methodologically to add analyses that are not in a protocol so long as these are disclosed as post hoc and they are justified (this happens in clinical trials and systematic reviews not infrequently). I also think it is completely fair for an author to decline to make changes requested by a reviewer because the author is not interested in the question(s) that these changes address, they do not have the time/resources, etc. And I agree that what is not ok is being asked to make post-hoc changes and then describe them as planned a priori, especially when selectively reporting information based on the nature of results (rather than on methodological rigor). In short, “selective reporting” is not allowed, but that is different from adding post hoc ancillary analyses to a paper. For example, I still think there is a missed opportunity not to conduct a formal qualitative analysis and discuss the nuances of differences in the reporting items identified in included studies that the authors grouped under the same “ACCORD item” in the data extraction form, though I understand the authors’ rationale that this was not something that they set out to do in their protocol. 1. The authors and I respectfully disagree about the need for a stand-alone limitations section in the narrative itself, which IME is standard practice in empirical articles at health journals. It would be helpful for the editor to weigh-in here, as the authors provide the following as a justification for not including such a section: “the journal provides a separate section about the strengths and limitations after the Abstract.” 2. Of the four articles that I suggested including in my last review, the authors wrote: “The first 2 did not meet our eligibility criteria as we explain, the 3rd was not indexed in any of the searched databases.” These reasons are limitations in their *search strategy* (screening procedures, databases), rather than the articles not meeting their *eligibility criteria*. As such, I still disagree with their exclusion.
--

	2a. I am surprised that the authors saw the title of the Sinha article and did not decide to check the full-text to see whether it discusses reporting (“recommendations for the future” in the title is how I first discovered this article myself years ago when trying to find research on how to both conduct and report Delphi studies). I’m also not sure what the authors obtained as an “abstract” in Rayyan, but one of the “Main Points” at the top of the article explicitly addresses reporting: “Methodological decisions should be CLEARLY DESCRIBED in the main publication in order to enable appraisal of the study.” The authors even note that “When one opens the full text, the study objectives declares: “methodological considerations or REPORTING for studies using the Delphi technique...”, but they then strangely claim that this “seems to be focusing on methodology of Delphi for COS, not on the reporting quality of the papers.” I am simply confused about this interpretation, as the word “reporting” is literally in that sentence. And then the authors state “this paper did include ... a few comments on reporting, as the authors found out that reporting was poor. This finding, though, was incidental and not the objective of their review.” Again, I am confused about this interpretation, as Table 1 is literally about the “REPORTING quality of the 15 included studies” and Table 2 is called “Recommended checklist that should be REPORTED in studies using the Delphi technique to determine which outcomes to measure in clinical trials or systematic reviews.” Moreover, these are the *only 2 tables* in the manuscript, and they both focus solely on reporting. To me, that the only two tables are about reporting indicates that reporting is not incidental, but rather is the primary objective of this paper. As such, I think it a limitation of the *search/screening methods* to have missed this study. That the authors also state that “The Sinha et al. paper and its insights will be considered by the Steering Committee, along with over 40 other papers, to draft the initial list of the ACCORD reporting guideline” indicate to me that they agree this article meets eligibility criteria. As such, excluding it is an omission. And it is not uncommon for systematic reviews to include such missing studies meeting eligibility criteria that are identified through peer reviewer. 2b. I have a similar reaction to their reasoning for continued exclusion of the Iglesias article. The authors state “The title “Reporting Guidelines for the Use of Expert Judgement in Model-Based Economic Evaluations” did not suggest directly that it aimed to improve health outcomes in biomedicine or clinical practice (nor did the abstract).” Yet this article has “reporting guidelines” and “expert judgement” in the title, the first sentence of the abstract clarifies that the topic is “healthcare interventions”, and the journal title indicates that they publish healthcare-related research (“PHARMACoeconomics”). As such, I am surprised the authors saw this information during screening and did not decide to check the full-text. That the authors also state “The reviewer is correct in saying that the paper by Iglesias et al. contains useful elements for the ACCORD reporting guideline, and by reading the full text we can see a rich discussion about the use of the terms “expert opinion” versus “expert elicitation”. The paper will be considered at a later stage of ACCORD development, albeit not included in this systematic review” indicates to me that they agree this article meets eligibility criteria. As such, excluding it is another omission. 2c. As the authors note, their search strategy missed the Toma paper. That the authors write “as stated above for the other studies
--	--

	that were not included, we thank the reviewer for having pointed them out and we will examine these papers for the ACCORD project development” indicates to me that they agree this article (and, again, the others above) meets eligibility criteria. As such, excluding it is another omission. They also state “we cannot include it retrospectively now.” Methodologically, I disagree: it is allowable to add eligible studies identified by peer reviewers that a search strategy failed to find. If the authors do not have the time/resources to add this (and the other two studies), that is a different reason, which should at the very least be explicitly stated in the manuscript (e.g., “one limitation of this study is that several eligible studies [INSERT REFS] were not identified by the search strategy.”)
--	---

VERSION 3 – AUTHOR RESPONSE

Reviewer 2

Comments to the Author:

I thank the authors for their attentiveness to reviewer comments in this revision to the paper. I found most responses (e.g., including more information on the data extraction form) to be helpful and improve the manuscript. I offer some remaining thoughts on the revised paper for the authors and editors to consider.

RESPONSE: Thank you for your kind words and additional suggestions.

Aside: As context for the below, I would like to clarify that I trust in specific future stages of ACCORD and the project overall to include the results reported in this manuscript and other findings of the project. That said, I do still think it important to critically appraise this review as a stand-alone piece, and thus that limitations should be addressed or stated explicitly in this paper (even if these limitations are planned to be addressed in future stages of the project). And after this second revision, I believe that there are some methodological issues where the authors and I respectfully though fundamentally disagree, in which case the editor will need to make a decision. Namely, one underlying issue is that the authors and I appear to disagree on the permissibility of retrospectively adding analyses. In general, I think it allowable methodologically to add analyses that are not in a protocol so long as these are disclosed as post hoc and they are justified (this happens in clinical trials and systematic reviews not infrequently). I also think it is completely fair for an author to decline to make changes requested by a reviewer because the author is not interested in the question(s) that these changes address, they do not have the time/resources, etc. And I agree that what is not ok is being asked to make post-hoc changes and then describe them as planned a priori, especially when selectively reporting information based on the nature of results (rather than on methodological rigor). In short, “selective reporting” is not allowed, but that is different from adding post hoc ancillary analyses to a paper. For example, I still think there is a missed opportunity not to conduct a formal qualitative analysis and discuss the nuances of differences in the reporting items identified in included studies that the authors grouped under the same “ACCORD item” in the data extraction form, though I understand the authors’ rationale that this was not something that they set out to do in their protocol.

1. The authors and I respectfully disagree about the need for a stand-alone limitations section in the narrative itself, which IME is standard practice in empirical articles at health journals. It would be helpful for the editor to weigh-in here, as the authors provide the following as a justification for not including such a section: “the journal provides a separate section about the strengths and limitations after the Abstract.” [NOTE FROM THE EDITORS: Our standard article format does not require a

standalone limitations section in the main text - instead we request that all limitations, including those highlighted during peer-review, are fully discussed in the Discussion section of the main text (in addition to the key limitations being highlighted in the 'Strengths and limitations of this study' bullet points after the abstract). Typically there would be a paragraph on limitations within the Discussion text, but so long as the limitations are adequately discussed within the Discussion, we are not strict about the specific format]

RESPONSE: The limitations of the study are, in accordance with the author guidelines of BMJOpen, not in a separate section, but intertwined in the whole discussion. We did add several new limitations to the discussions in the section throughout. These will be addressed below.

2. Of the four articles that I suggested including in my last review, the authors wrote: "The first 2 did not meet our eligibility criteria as we explain, the 3rd was not indexed in any of the searched databases." These reasons are limitations in their *search strategy* (screening procedures, databases), rather than the articles not meeting their *eligibility criteria*. As such, I still disagree with their exclusion.

2a. I am surprised that the authors saw the title of the Sinha article and did not decide to check the full-text to see whether it discusses reporting ("recommendations for the future" in the title is how I first discovered this article myself years ago when trying to find research on how to both conduct and report Delphi studies). I'm also not sure what the authors obtained as an "abstract" in Rayyan, but one of the "Main Points" at the top of the article explicitly addresses reporting: "Methodological decisions should be CLEARLY DESCRIBED in the main publication in order to enable appraisal of the study." The authors even note that "When one opens the full text, the study objectives declares: "methodological considerations or REPORTING for studies using the Delphi technique...", but they then strangely claim that this "seems to be focusing on methodology of Delphi for COS, not on the reporting quality of the papers." I am simply confused about this interpretation, as the word "reporting" is literally in that sentence. And then the authors state "this paper did include ... a few comments on reporting, as the authors found out that reporting was poor. This finding, though, was incidental and not the objective of their review." Again, I am confused about this interpretation, as Table 1 is literally about the "REPORTING quality of the 15 included studies" and Table 2 is called "Recommended checklist that should be REPORTED in studies using the Delphi technique to determine which outcomes to measure in clinical trials or systematic reviews." Moreover, these are the *only 2 tables* in the manuscript, and they both focus solely on reporting. To me, that the only two tables are about reporting indicates that reporting is not incidental, but rather is the primary objective of this paper. As such, I think it a limitation of the *search/screening methods* to have missed this study. That the authors also state that "The Sinha et al. paper and its insights will be considered by the Steering Committee, along with over 40 other papers, to draft the initial list of the ACCORD reporting guideline" indicate to me that they agree this article meets eligibility criteria. As such, excluding it is an omission. And it is not uncommon for systematic reviews to include such missing studies meeting eligibility criteria that are identified through peer reviewer.

RESPONSE: It is common practice to screen for systematic reviews based on title and abstract. The abstract uploaded in Rayyan is the same abstract as can be seen in PubMed <https://pubmed.ncbi.nlm.nih.gov/21283604>. We added the abstract in our former response. Screening has been done by two people independently. The title suggests it is about choosing outcomes in clinical trials, and the abstract states that it is about developing core outcome sets. We don't agree this is a limitation in our search strategy or screening process, it is rather a limitation in the reporting of the article. As in a normal systematic review, it is not possible to read all references in full (that is why we screen based on title and abstract). We have piloted 4 search strategies before we ended up with the last search. However, we have added several lines in the discussion about the studies that were brought to our attention during the peer review process and provided some details as why we did not include them.

2b. I have a similar reaction to their reasoning for continued exclusion of the Iglesias article. The authors state “The title “Reporting Guidelines for the Use of Expert Judgement in Model-Based Economic Evaluations” did not suggest directly that it aimed to improve health outcomes in biomedicine or clinical practice (nor did the abstract).” Yet this article has “reporting guidelines” and “expert judgement” in the title, the first sentence of the abstract clarifies that the topic is “healthcare interventions”, and the journal title indicates that they publish healthcare-related research (“PHARMACOEconomics”). As such, I am surprised the authors saw this information during screening and did not decide to check the full-text. That the authors also state “The reviewer is correct in saying that the paper by Iglesias et al. contains useful elements for the ACCORD reporting guideline, and by reading the full text we can see a rich discussion about the use of the terms “expert opinion” versus “expert elicitation”. The paper will be considered at a later stage of ACCORD development, albeit not included in this systematic review” indicates to me that they agree this article meets eligibility criteria. As such, excluding it is another omission.

RESPONSE: The study by Iglesias et al. does not aim to improve health outcomes, but to evaluate health economics. Pharmacoeconomics is outside the scope of our study and the article contains points already made in our paper. We have added a line in the discussion section stating that the article identified similar gaps but that its scope is outside our protocol and research question.

2c. As the authors note, their search strategy missed the Toma paper. That the authors write “as stated above for the other studies that were not included, we thank the reviewer for having pointed them out and we will examine these papers for the ACCORD project development” indicates to me that they agree this article (and, again, the others above) meets eligibility criteria. As such, excluding it is another omission. They also state “we cannot include it retrospectively now.” Methodologically, I disagree: it is allowable to add eligible studies identified by peer reviewers that a search strategy failed to find. If the authors do not have the time/resources to add this (and the other two studies), that is a different reason, which should at the very least be explicitly stated in the manuscript (e.g., “one limitation of this study is that several eligible studies [INSERT REFS] were not identified by the search strategy.”).

RESPONSE: As stated in our methods, the databases (9) searched for the systematic review are Embase, MEDLINE, Web of Science, MEDLINE (web of science), PubMed, Cochrane Library, Emcare, PsychINFO and Academic Search. This paper by Toma et al. could only have been found by Google Scholar and therefore is ineligible for inclusion. However, we disclose this point in the discussion section of the manuscript. Of note, the study by Toma et al. was published in a journal with all the characteristics of a predatory journal, so we could not trust it was peer-reviewed (this was also confirmed by our information specialist).